# Burden of mortality and its predictors among TB-HIV co-infected patients in Ethiopia: Systematic review and meta-analysis

Amare Kassaw[1]*, Demewoz Kefale[1], Tigabu Munye Aytenew[2], Molla Azmeraw[3], Muluken Chanie Agimas[4], Shegaw Zeleke[2], Mastewal Ayehu Sinshaw[5], Nigatu Dessalegn[6], Worku Necho Asferie[7]

1 Department of Pediatrics and Child Health Nursing, College of Health Sciences, Debre Tabor University, Debre Tabor, Ethiopia, 2 Department of Adult Health Nursing, College of Health Sciences, Debre Tabor University, Debre Tabor, Ethiopia, 3 Department of Pediatrics and Child Health Nursing, College of Health Sciences, Woldia University, Woldia, Ethiopia, 4 Department of Epidemiology and Biostatics, Institute of Public Health, College of Medicine and Health Sciences, University of Gondar, Gondar, Ethiopia, 5 Department of Nursing, Tibebe Gion Specialized Hospital, Bahir Bar University, Bahir Bar University, Bahir Dar, Ethiopia, 6 Department of Pediatrics and Child Health Nursing, College of Health Sciences, Injibara University, Injibara, Ethiopia, 7 Department of Maternal and Neonatal Health Nursing, College of Health Sciences, Debre Tabor University, Debre Tabor, Ethiopia

* amarekassaw2009@gmail.com

## Abstract

### Background

Human immunodeficiency virus (HIV) and tuberculosis (TB) are still the two major deadly pandemics globally, causes 167,000 deaths in 2022. The two lethal combinations pose a substantial challenge to public health, especially in areas with high burden of both diseases such as Sub-Saharan Africa including Ethiopia. However, there is no study that showed national figure on mortality of TB/HIV co-infected patients. Hence, this review intended to provide pooled mortality rate and its predictors among patients co-infected with twin pandemics.

### Methods

Using reputable electronic data bases, primary studies were searched from January 25 to February 5, 2024. The review included papers published in English language conducted between 2004 and 2024. Heterogeneity between included studies was evaluated using Cochrane Q-test and the $I^2$ statistics. Sub-group analysis was done to mitigate significant heterogeneity. Sensitivity analysis was also done to evaluate the effect of single studies on pooled estimated result.

### Results

In this systematic review and meta-analysis a total of 5,210 study participants were included from 15 primary studies. The review disclosed that the pooled proportion and incidence of mortality were 18.73% (95% CI: 15.92-20.83) and 4.94 (95% CI: 2.98-6.89) respectively. Being bedridden and ambulatory functional status, poor ART adherence, CD4 count below

**Data Availability Statement:** All relevant data are within the article and its supporting information files.

**Funding:** The author(s) received no specific funding for this work.

**Competing interests:** The authors have declared that no competing interests exist.

**Abbreviations:** HIV, Humane Immune Virus; AIDS, Acquired Immune Deficiency Virus; HAART, Highly Active Antiretroviral Therapy; ART, Antiretroviral Therapy; AHR, Adjusted Hazard Ratio; CI, Confidence Interval; TB, Tuberculosis; PTB, Pulmonary tuberculosis; IPT, Isoniazid preventive therapy; CPT, Cotrimoxazole preventive therapy; WHO, World Health Organization.

the threshold **(<**200 cells/mm$^3$), advanced WHO clinical staging, not provision of cotrimoxazole and isoniazid preventing therapy, anemia and extra pulmonary TB were significant predictors of mortality.

## Conclusion and recommendations

The analyzed data of this systematic review and meta-analysis depicted that the national pooled proportion and incidence of mortality among TB-HIV co-infected patients were considered to be still high. The authors strongly recommended scale up and continuous provision of cotrimoxazole and isoniazid preventive therapy. In addition, early identification and treatment of anemia will greatly halt the high burden of mortality. Generally, to reduce mortality and improve survival, a collaborative effort is mandatory to emphasize close follow up of patients with identified predictors.

## Introduction

Human immunodeficiency virus (HIV) and tuberculosis (TB) are the two major deadly pandemics worldwide [1–4]. According to World Health Organization (WHO) global report, about 214,000 in 2020 and 187,000 people in 2021 died due to TB-HIV co-infections [1, 5]. The organization also reported that the two lethal combinations have been the cause of 167,000 deaths in 2022. In the absence of appropriate treatment, nearly all TB-HIV co-infected patients will not survive [6]. Evidence showed that co-infection with HIV and TB accounted for roughly 40% of hospital deaths [7].

Both diseases are interacting in a complex and synergetic manner to influence one on other that leads to rapidly decrease the patients' immunity system [8]. TB and HIV co- infection poses a substantial challenge to public health, especially in areas with high burden of both diseases such as Sub-Saharan Africa including Ethiopia [9]. HIV- infection weakens the immune system and makes the patient to be susceptible for tuberculosis infection and increase the progress from latent TB infection to active TB infection [10]. This weakened immune system gives a good opportunity for tuberculosis to progress rapidly and leads to more severe disease and increased risk of mortality and low survival [11]. TB infection also hastens the progress of HIV-infection by triggering an immune response that can increase HIV replication and viral load [10, 12]. The activation of immunity due to tuberculosis can induce inflammation and immune system dysfunction that alter the effectiveness of antiretroviral therapy [13].

Burden of mortality rate among TB-HIV co-infected patients is very high compared to having either disease alone [14, 15]. According to WHO, in 2017 the two pandemics attribute to the death of 300,000 people globally [16]. A finding from worldwide systematic review and meta-analysis revealed that the mortality of adults attributable to tuberculosis was 24.9% and 30.1% in children respectively [7]. The study conducted in china disclosed that the mortality rate among TB-HIV co-infected patients was 5.3% [17]. A similar study from China also showed that the mortality of patients co-infected with TB and HIV was 34.7% [18]. A cluster randomized trial study conducted in South Africa indicated that a mortality rate among TB-HIV co-infected patients was 10.1 per 100 person-years [19]. A study in Malaysia also depicted that the mortality of TB patients co-infected with HIV was 23.3% [20]. Evidence in Tanzania disclosed that the incidence of mortality among TB-HIV patients was 57.8 per 1000

person-years [11]. The risk of mortality is also high in Ethiopia among patients co-infected with these jeopardize infectious disease [21].

Several predicting factors were contributing for high mortality rate of patients with TB/HIV co- morbidities. From previous studies, the attributing factors of mortality among TB-HIV infected patients were low CD4 count [18, 22], WHO clinical stage 3 and 4 [23–25], ambulatory and bedridden functional status [26], being male gender [27], age of patients [28], residency [29, 30], missing of cotrimoxazole preventive therapy [30, 31] and not disclosing HIV status [32].

The globe has taken several measures to reduce the burden of morbidity and mortality due to TB-HIV co-infections as well as to eradicate in the near future. Joint United Nations Programme on HIV/AIDS (UNAIDS) is striving to end these two intertwined epidemics by 2030 through Fast Track and End TB Strategies [33]. TB/HIV collaborative activities undergone in in Latin America and the Caribbean countries have been scaled up and showed a tendency toward improvements [34]. Africa as a continent has straggled by setting Abuja declarations to end the pandemics by 2030 [35]. Ethiopia on its part has planned and implemented remarkable measures to tackle these catastrophic pandemics and to save million lives [21, 36]. Despite these interventions and successful achievements, TB/HIV co- infection is still the leading cause of death, particularly in resource limited settings [37, 38].

In Ethiopia several studies were conducted to show the high burden and incidence of death among patients as a result of TB-HIV co-infection and that indicated the mortality rate ranges from 12.3% in Southern Nation, Nationalities and Peoples Region (SNNPR) [39] and 35.39 in Amhara region [8]. The highest and the lowest incidence rate of mortality also found in Oromia [40] (14.56/100 person-years) and Tigray [41] (1.7/100 person-years) regions respectively. The finding showed that there is inconsistency and inconclusiveness among studies in different regions of the country. Therefore, the objective of this review was to determine the national level pooled mortality rate and its predictors among primary studies conducted on mortality of TB-HIV co- infected patients in Ethiopia.

## Methods

### Reporting

This review was done according to the preferred reporting systematic review and meta-analysis (PRISMA) guideline (S1 Checklist), and prospectively registered at the Prospero with a registration number of CRD42024511756.

### Search strategy and study selection

Primary studies conducted in English were retrieved using international reputable electronic data bases. HINARI, PubMed, CINHAL, Scopus, web of science, Google scholar, Embase and Cochrane Library were systematically searched from January 25 to February 5, 2024. Open Google, university repositories and reference lists of eligible articles were also searched to find eligible articles. The eligible studies were accessed using the following search terms, keywords and Medical Subject Headings (MeSH terms): "Incidence"[Mesh], "incidence rate", "incidence density", magnitude, "Survival"[Mesh], "Mortality"[Mesh], predictors, "TB-HIV co-infection", "Child"[Mesh], "Adult"[Mesh], "Persons"[Mesh], "Adolescent"[Mesh] and Ethiopia. The search terms and key words were combined using "AND" and "OR" Boolean operators to yield sufficient and appropriate search results (S1 Table). To access the eligible articles, the search was guided by the adapted PICO framework.

## Eligibility criteria of studies

The search results of electronic search engine were exported to Endnote X8 software. Two authors (WNA and DK) were assigned to remove unrelated studies based on their title and abstract. They also screened full text articles according to pre-determined inclusion and exclusion criteria. Any arguments between the authors about eligibility of articles were disputed through discussion and interference of other reviewer members.

## Inclusion and exclusion criteria

The authors have included studies conducted on mortality and its predictors among TB-HIV co-infected patients in Ethiopia. The review incorporated articles published in English language conducted between 2004 and 2024 since TB/HIV collaborative activities were piloted in 2004.Papers which do not report outcome interest and without full text articles were excluded. Additionally, studies conducted on multi-drug or extensively drug resistant tuberculosis and HIV co-infection as outcome interest were excluded (S2 Table).

## Outcome measurement

The main outcome of this systematic review and meta-analysis was pooled proportion mortality rate among TB/ HIV co - infected patients. Pooled incidence mortality rate and its predictors were also other outcomes of the review.

## Data extraction and screening process

AK and SZ screened the articles titles and abstracts against the inclusion and exclusion criteria. Then, relevant data were extracted by two reviewers (MA and TMA) according to predetermined exclusion and inclusion criteria from January 7 to 13/ 2024. Discrepancies between the authors were settled by discussion. First author name, publication year, study region, study design, study setting, sample size, outcome, response rate, incidence rate of mortality, mortality rate, predictors' effect size (AHR) were extracted on Microsoft excel spread sheet (S1 Data).

## Handling of missing data

The missing data were handled through complete case analysis method. There is no evidence to perform single or multiple imputation since the missing value is less than 5%.

## Quality assessment

The Joanna Briggs Institute (JBI) quality assessment checklists for cohort studies were used to evaluate the quality of included studies. MCA and MAS were assigned to critical appraisal of the articles and disagreement between were resolved by other reviewers. The tool has the following criteria: appropriate statistical analysis used strategies to address incomplete follow up utilized, sufficient follow up time, measurement of the outcomes in a valid and reliable way, participants free of outcome at the beginning of the study, identifying confounding and strategies to reduce it. Accordingly, questions that fulfill the above criteria labeled as 1 and 0 for question that did not fulfill the criteria. Studies were considered to be low risk/high quality when scored 50% or higher on the quality assessment tools. Whereas, studies scored less than 49% on quality assessment checklist were categorized as high risk/low quality [42]. Therefore, in this systematic and met analysis all articles were scored greater than 50% (S2 Checklist).

## Statistical analysis

After the data were extracted on Microsoft excel, exported to STATA version 17 statistical software for further analysis. The pooled mortality rate among TB- HIV co-infected patients and its predictors was estimated using random effects model using DerSimonian-Laird model weight. The presence of heterogeneity between included studies was evaluated using Cochrane Q-test and the $I^2$ statistics. Publication bias was checked through graphical (funnel plot) and statistical (Egger's) test. Sub-group analysis was done to adjust random variation in the presence of significant heterogeneity between primary studies. A leave-one –out sensitivity analysis was done to evaluate the effect of single studies on pooled estimated result.

## Operational definitions

**Pooled proportion of mortality.** pooled proportion of mortality is the ratio of the total death of patients co- infected with TB and HIV and the total case of patients infected with TB-HIV followed for a specified period of time.

**Pooled incidence rate of mortality.** The pooled incidence of mortality among TB-HIV co- infected patients is the mortality of an incident TB-HIV case, or event of interest which is the pooled result of mortality of new TB-HIV case during a given observation period/person-time observed.

**Ethical consideration.** Ethical clearance is not applicable for this systematic review and meta-analysis

# Results

The search results of combined electronic data bases yield a total of 1051 articles. Of which, 388 studies were removed because of duplication. After careful reading of the titles and abstracts, 501 articles were also not retrieved. Again, 103 studies were removed as they reported unmatched outcome interest. Additionally, 44 articles were excluded after reading full text articles since studies not in Ethiopia, systematic reviews and other reasons. Finally, 15 studies were included for analysis of systematic review and meta-analysis (Fig 1).

## Characteristics of included articles

In this systematic review and Mata analysis, fifteen primary studies [8, 26, 39–41, 43–52] with retrospective cohort study design were included to estimate the pooled mortality rate among patients with TB/HIV co –infection. Of these studies, only 10 articles [39–41, 43–45, 49–52] were used to determine the incidence of mortality rate. A total of 5,210 study participants were included with the smallest sample size (227) [44] from Amhara region and the maximum were from Oromia region (566) respectively [49]. Five studies were conducted in Oromia region [26, 40, 47, 48, 50], four in Amhara region [8, 43, 44, 49], three in SNNPRS [39, 45, 52] and 2 studies were done in Tigray region [41, 46] (Table 1).

## Pooled proportion mortality rate of TB/HIV co-infected patients in Ethiopia

A total of 15 studies [8, 26, 39–41, 43–52] were used to estimate the pooled proportion mortality rate of patients co-infected with TB/HIV. Based on the random effects model, the mortality rate was 18.73% (95% CI: 15.92-20.83) with significant heterogeneity (I2=82.12%) (Fig 2).

Ten studies were also used to estimate the pooled incidence mortality rate among TB-HIV co-infected patients. Accordingly, the overall pooled incidence mortality rate was 4.94 (95% CI: 2.98-6.89) with substantial heterogeneity ($I^2$= 88.72%) (Fig 3).

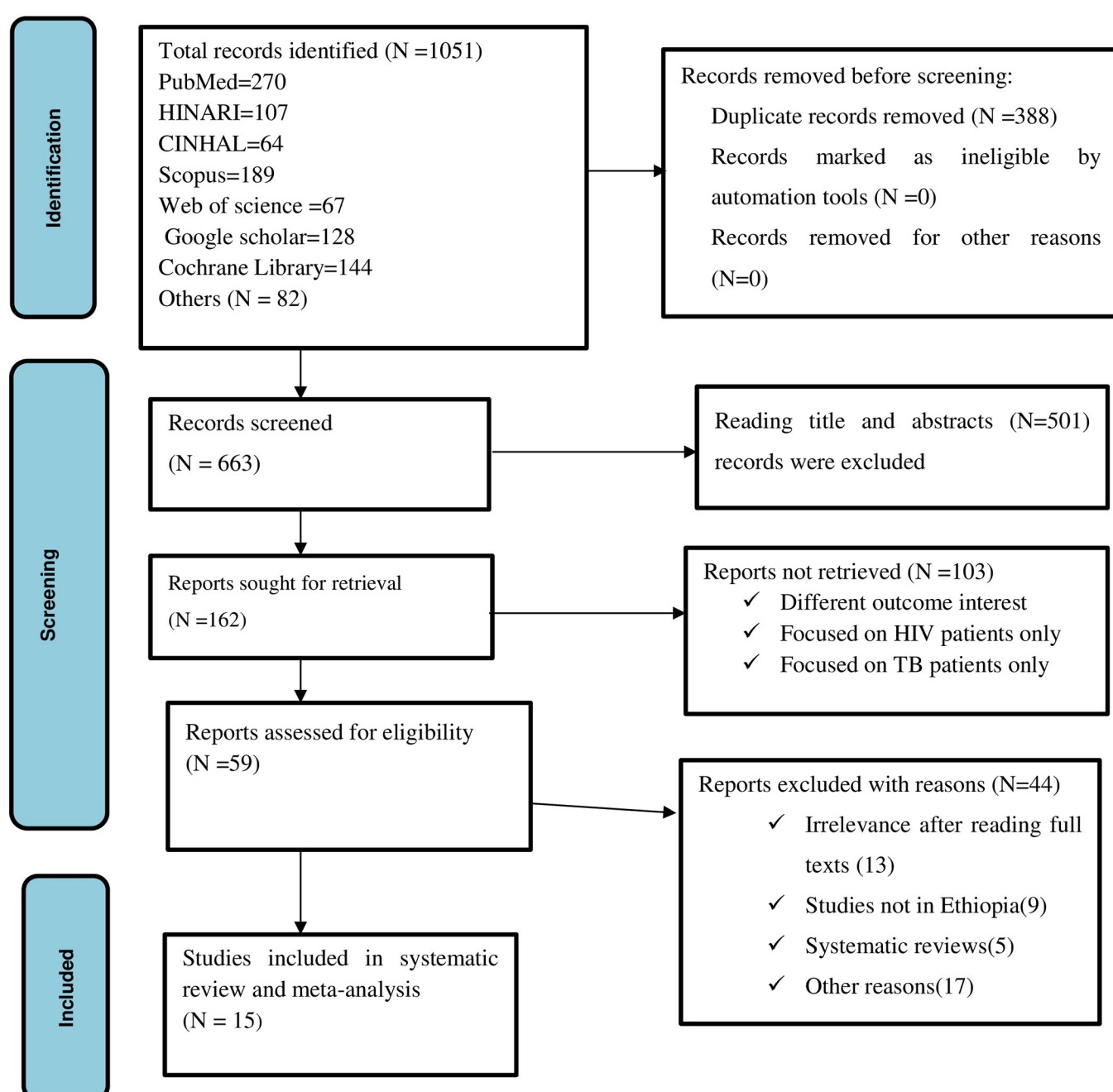

**Fig 1. PRISMA2020 flow diagram of article selection for systematic review and meta-analysis of mortality among TB-HIV co-infected patients in Ethiopia.**

**Table 1. Characteristics of included studies among TB/HIV co-infected patients with outcomes.**

| First authors with publication year | Study region | Study design | Sample size | Mortality rate | IR/100PYO | Study quality |
|---|---|---|---|---|---|---|
| Atalel et al., 2018 [43] | Amhara | Retrosp. cohort | 271 | 14.02 | 3.27 | Low risk |
| Birhan et al., 2021 [8] | Amhara | Retrosp. cohort | 243 | 35.39 | | Low risk |
| Chanie et al., 2021 [44] | Amhara | Retrosp. cohort | 227 | 17.18 | 3.7 | Low risk |
| Dawit et al., 2021 [45] | SNNPRS | Retrosp. cohort | 274 | 17.15 | 2.97 | Low risk |
| Gemechu et al., 2022 [39] | SNNPRS | Retrosp. cohort | 284 | 12.3 | 2.78 | Low risk |
| Gesesew et al., 2016 [26] | Oromia | Retrosp. cohort | 272 | 20.2 | | Low risk |
| Gezea et al., 2020 [46] | Tigray | Retrosp. cohort | 305 | 23 | | Low risk |
| Habtegiorgis et al., 2023 [47] | Oromia | Retrosp. cohort | 471 | 16.8 | | Low risk |
| Lelisho et al., 2022 [40] | Oromia | Retrosp. cohort | 402 | 20.9 | 14.56 | Low risk |
| Nigussie et al., 2021 [41] | Tigray | Retrosp. cohort | 253 | 15 | 1.7 | Low risk |
| Refera et al., 2013 [48] | Oromia | Retrosp. cohort | 501 | 15.8 | | Low risk |
| Silesh et al., 2013 [49] | Amhara | Retrosp. cohort | 422 | 22.04 | 4.09 | Low risk |
| Sime et al., 2022 [50] | Oromia | Retrosp. cohort | 566 | 13.4 | 11.04 | Low risk |
| Teklu et al., 2017 [51] | Ethiopia | Retrosp. cohort | 355 | 14 | 2.5 | Low risk |
| Wondimu et al., 2020 [52] | SNNPRS | Retrosp. cohort | 364 | 22.8 | 5.02 | Low risk |

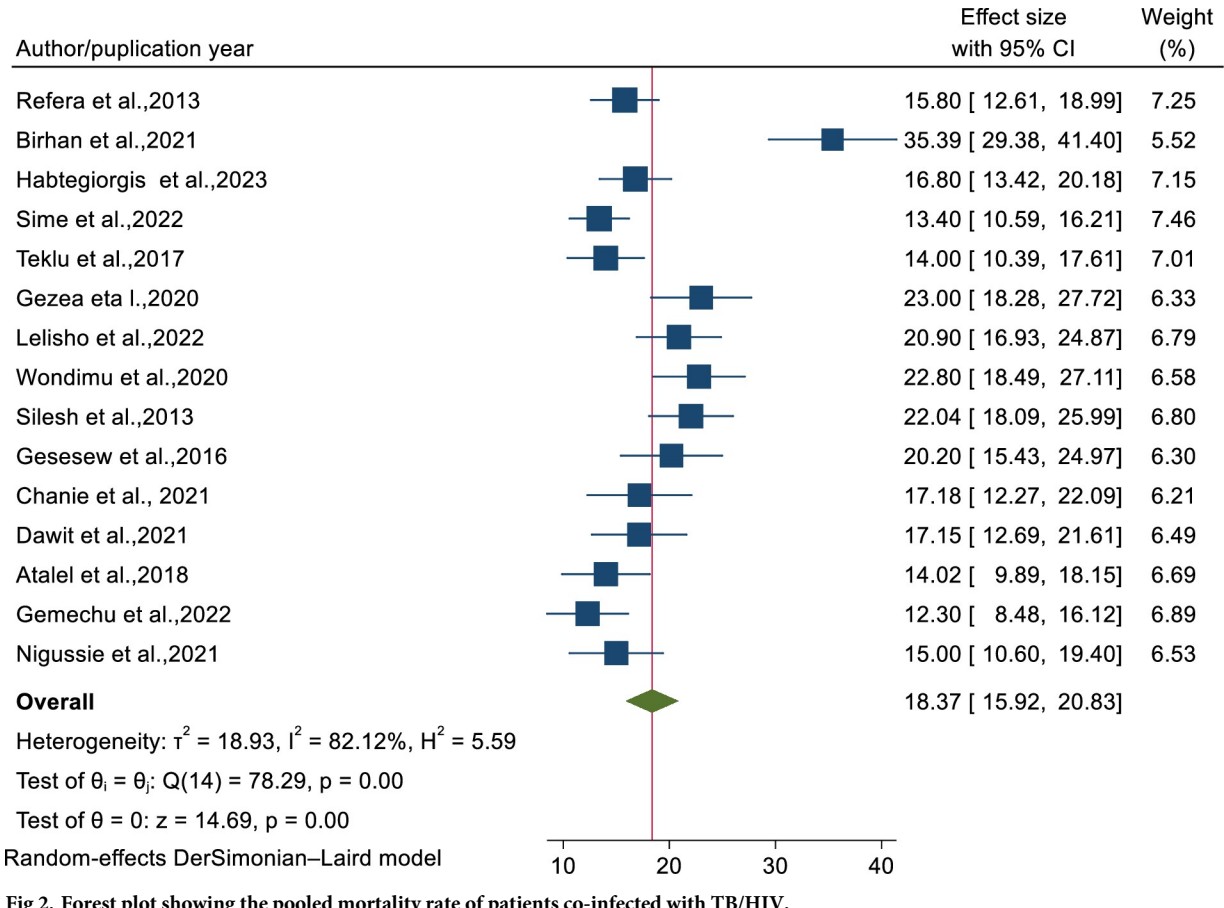

**Fig 2. Forest plot showing the pooled mortality rate of patients co-infected with TB/HIV.**

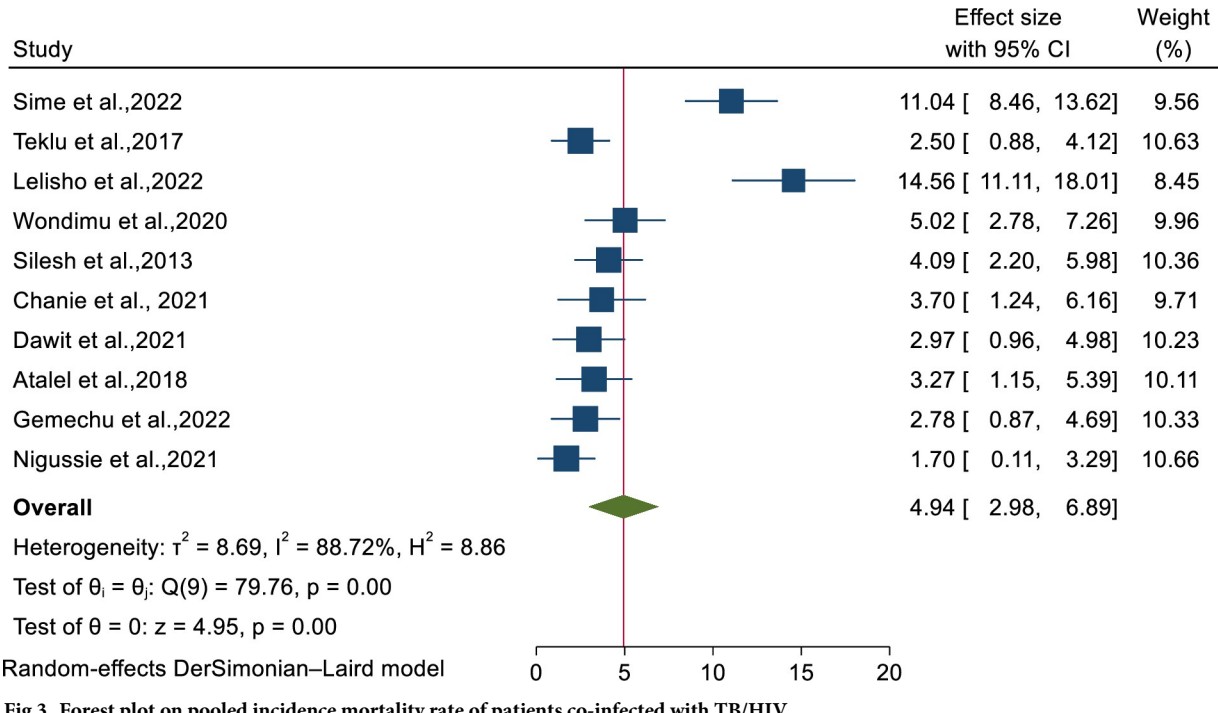

**Fig 3. Forest plot on pooled incidence mortality rate of patients co-infected with TB/HIV.**

## Handling heterogeneity

To handle the observed heterogeneity, sub-group and sensitivity analysis were carried out on mortality rate and incidence density mortality rate among TB/HIV co-infected patients. Sub-group analysis was performed using publication year, region, sample size and age of patients. Based on this, mortality of patients was high among studies conducted after 2020 as compared to studies done before 2020 (Fig 4). Sub-group analysis concerning to region, the highest proportion of mortality rate was found in Amhara region followed by Tigray regional state (S1 Fig). Mortality rate among patients age greater than 15 years (adults) was higher as compared to their counterparts (S2 Fig). Similarly, mortality rate among TB/HIV co-infected patients with sample size less than 350 was higher compared to studies with sample size greater than 350 (Fig 5).

Regarding to sub-group analysis of incidence of mortality rate, studies conducted exclusively on adult patients has higher incidence of mortality rate than studies done on children (S3 Fig). Based on the region, the highest incidence of mortality was also observed in Oromia region followed by Amhara region (S4 Fig). Likewise, the new occurrence of mortality was greater among studies published after the year 2020 than before 2020 (S5 Fig).

## Sensitivity analysis and publication bias

The result of leave-one out sensitivity analysis showed that there is no single study that excessively affects the pooled estimated effect of mortality among TB-HIV co-infected patients (Fig 6). To identity publication bias, we performed statistical test (Eggers' test) and visual inspection (funnel plot). The graphical presentation of funnel plot showed that there is symmetrical distribution of studies. Hence, this declares there is no small study effect (S6 Fig). Similarly, Eggers' statistical test analysis revealed that there is no publication bias since p-value is insignificant (p-value=0.74).

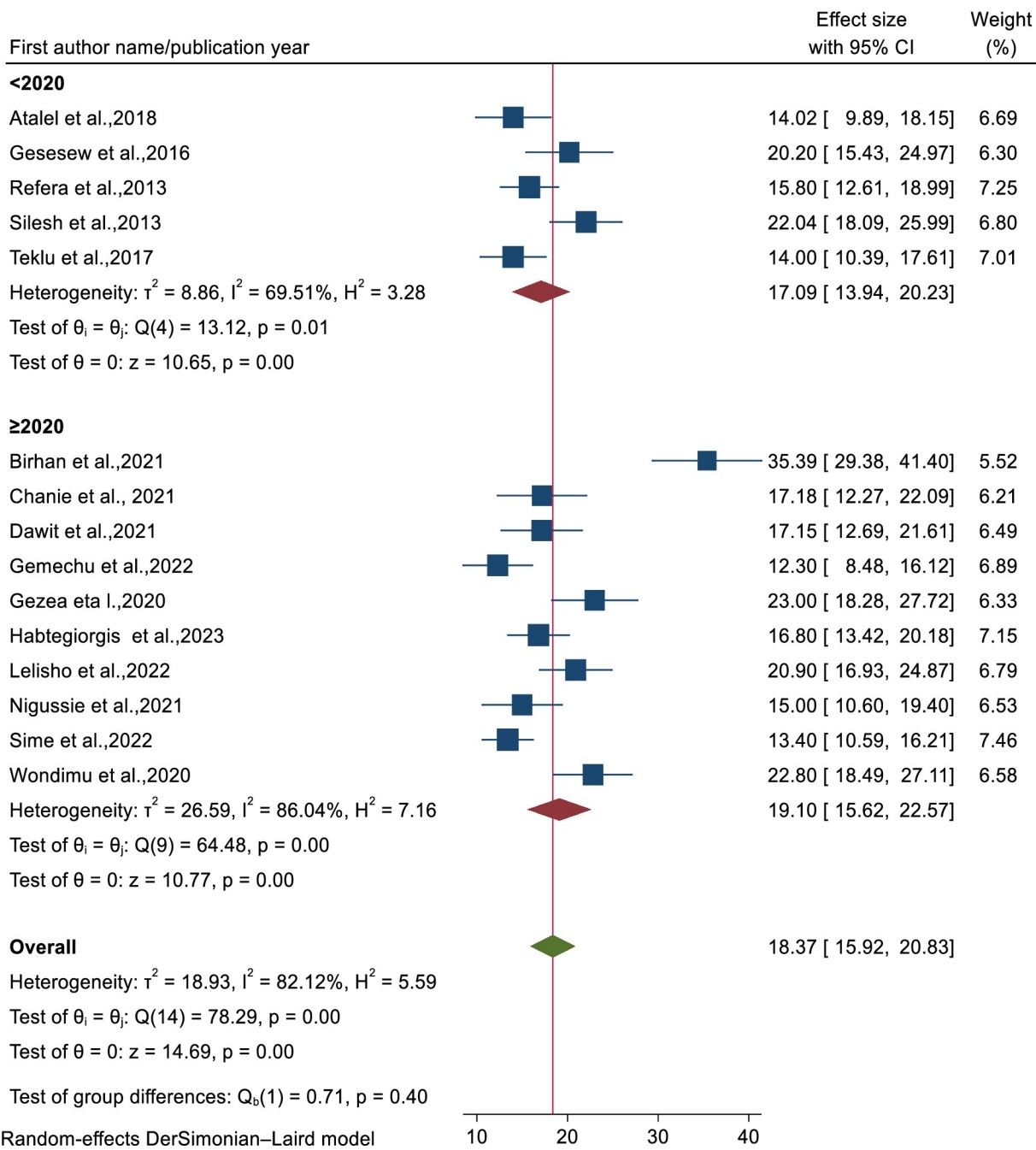

**Fig 4. Forest plot of sub-group analysis according to publication year on mortality rate among TB/HIV co-infected patients in Ethiopia.**

## Pooled effects of predictors on the mortality of patients co- infected with TB/HIV

We conducted meta- analysis to determine the pooled effects of predictors of mortality among TB/HIV co-infected patients. Bedridden and ambulatory functional status, poor ART adherence, CD4 count below the threshold ($<200$ cells/mm$^3$), advanced WHO clinical staging, missing of cotrimoxazole and isoniazid preventing therapy, anemia, and extra pulmonary TB were significant predictors of mortality (Table 2).

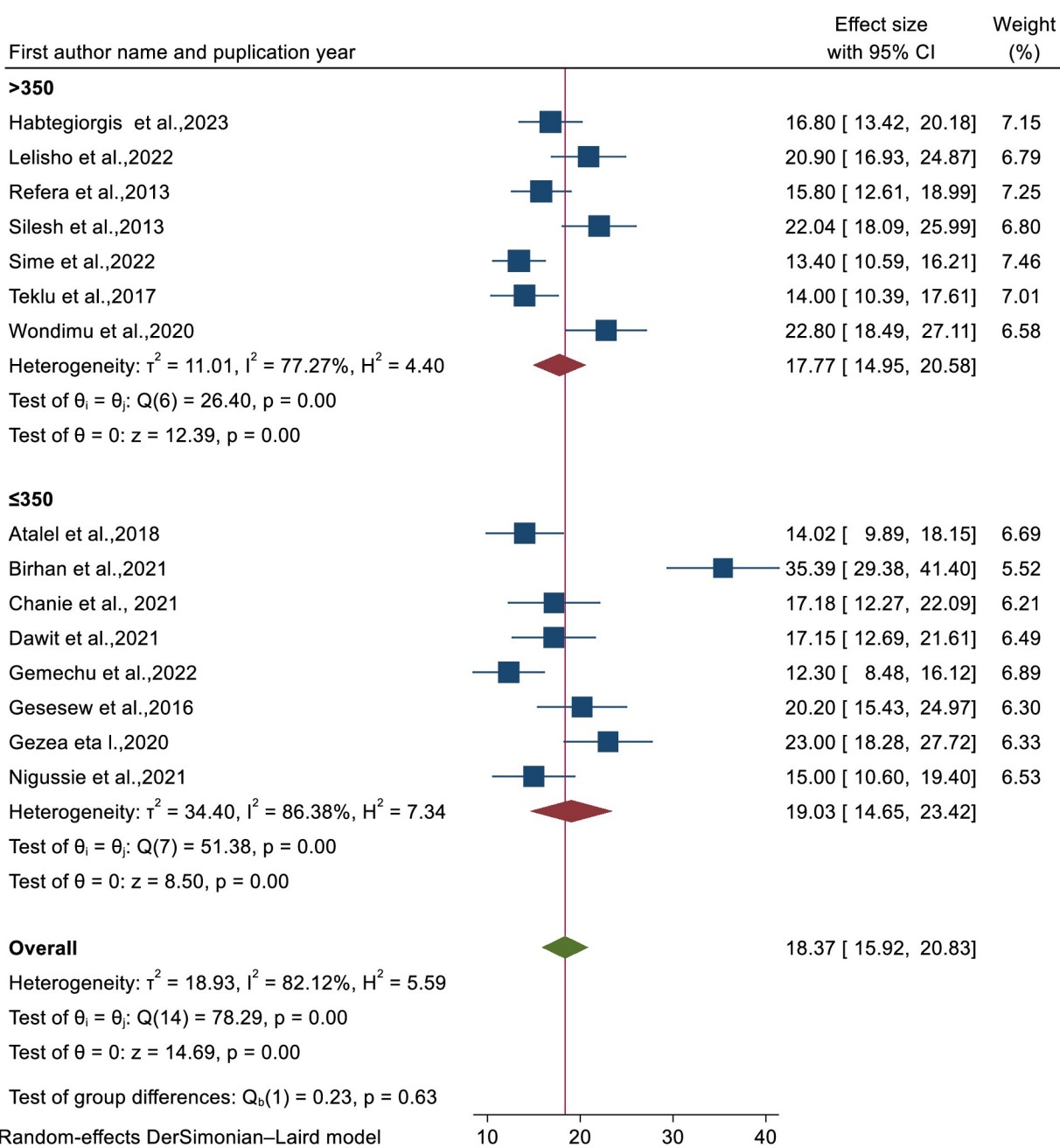

**Fig 5. Forest plot showing sub-group analysis based on sample size on mortality rate among TB/HIV co-infected patients in Ethiopia.**

Four studies [39, 41, 43, 45] were used to assess the association of poor ART adherence and mortality of patients co-infected with TB/HIV. A random effects model analysis indicated that patients with poor and fair ART adherence had 3.88 (AHR=3.88; 95%CI: 2.890-4.88) times hazard to death than good adherence. Similarly, patients with low CD4 count were 3.48 times (AHR=3.48; 95%CI: 1.74-5.22) more likely to die compared to their counterparts.

A total seven studies [40, 43, 44, 46, 47, 49, 52] were combined to determine the association of cotrimoxazole preventing therapy and mortality of TB/HIV co-infected patients. Significant heterogeneity were observed among studies ($I^2$=97.49%, p-value<0.001). Leave-one –out

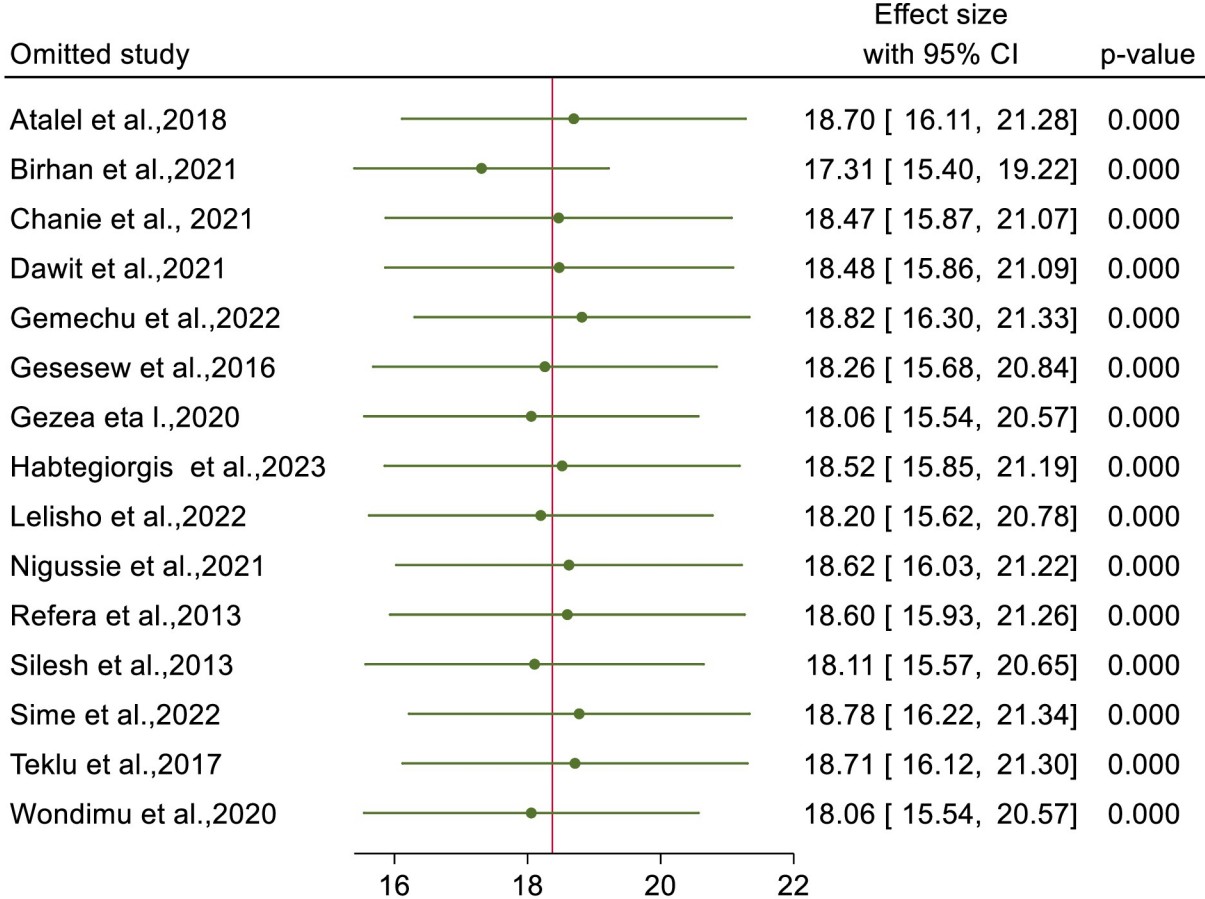

| Omitted study | | Effect size with 95% CI | p-value |
|---|---|---|---|
| Atalel et al.,2018 | | 18.70 [ 16.11, 21.28] | 0.000 |
| Birhan et al.,2021 | | 17.31 [ 15.40, 19.22] | 0.000 |
| Chanie et al., 2021 | | 18.47 [ 15.87, 21.07] | 0.000 |
| Dawit et al.,2021 | | 18.48 [ 15.86, 21.09] | 0.000 |
| Gemechu et al.,2022 | | 18.82 [ 16.30, 21.33] | 0.000 |
| Gesesew et al.,2016 | | 18.26 [ 15.68, 20.84] | 0.000 |
| Gezea eta l.,2020 | | 18.06 [ 15.54, 20.57] | 0.000 |
| Habtegiorgis et al.,2023 | | 18.52 [ 15.85, 21.19] | 0.000 |
| Lelisho et al.,2022 | | 18.20 [ 15.62, 20.78] | 0.000 |
| Nigussie et al.,2021 | | 18.62 [ 16.03, 21.22] | 0.000 |
| Refera et al.,2013 | | 18.60 [ 15.93, 21.26] | 0.000 |
| Silesh et al.,2013 | | 18.11 [ 15.57, 20.65] | 0.000 |
| Sime et al.,2022 | | 18.78 [ 16.22, 21.34] | 0.000 |
| Teklu et al.,2017 | | 18.71 [ 16.12, 21.30] | 0.000 |
| Wondimu et al.,2020 | | 18.06 [ 15.54, 20.57] | 0.000 |

Random-effects DerSimonian–Laird model

**Fig 6. Leave-one out sensitivity analysis of mortality among TB/HIV co-infected patients in Ethiopia.**

sensitivity analysis showed that there is no single study that affects the result of pooled estimated effect (S7 Fig). Egger's statistical test evidenced that there is no publication bias among studies (P-value=0.5411).

Bedridden patients had 3.30 times hazard to death (AHR=3.30: 95%CI: 2.07- 4.53) as compared to their counterparts with substantial heterogeneity among included studies ($I^2$=93.43%, P-value<0.001). The analysis results of random effects model disclosed that there is no a single study that excessively affect the overall estimated result (S8 Fig) and also there is no evidence of small study effect among studies (p-value= 0.6531).

Seven primary studies [26, 39–41, 47, 48, 50] were used to identify the pooled effects of association between WHO clinical stage IV and mortality of patients with TB/HIV co-infection. A random effects model analysis indicated that TB/HIV co-infected patients who had WHO stage IV were about 4 times hazard to death (AHR=4.17; 95%CI:3.28-5.06) than stage I and Stage II. There was significant heterogeneity among included studies ($I^2$=84.01%, p-value=0.001). From sensitivity analysis, no single study affects the pooled estimated finding (S9 Fig).

Likewise, eight studies [39, 41, 43, 45–49] were used to identify the pooled estimated effect of the association between extra pulmonary tuberculosis and mortality of TB/HIV co-infected patients. Patients with extra PTB were 2.56 (AHR=2.56; 95%CI: 1.36-3.76) times more likely to die than PTB patients. Leave-one –out sensitivity analysis showed that there is no a single

**Table 2. The pooled effects of predictors on mortality of TB/HIV-infected patients in Ethiopia.**

| Variables | Variable Categories | Included studies | AHR (95% CI) | Heterogeneity)I2%, P-value) | Egger's P-value |
|---|---|---|---|---|---|
| ART adherence | Fair &poor | 4 | 3.88(2.890-4.88) | 77.86, 0.001 | 0.8007 |
| | Good | | 1 | | |
| CD4 count | Below threshold | 5 | 3.48(1.74-5.22) | 97.49, 0.001 | 0.1971 |
| | Above threshold | | 1 | | |
| Taking CPT | No | 7 | 2.73(2.12-3.33) | 85.04, 0.001 | 0.5411 |
| | Yes | | 1 | | |
| Anemia | Yes | 3 | 2.80(2.23-3.37) | 27.08, 0.001 | 0.765 |
| | No | | 1 | | |
| Taking IPT | No | 4 | 3.10(2.72- 3.48) | 0.00, 0.001 | 0.9681 |
| | Yes | | 1 | | |
| Bedridden | Yes | 5 | 3.30(2.07- 4.53) | 93.43, 0.001 | 0.6531 |
| | No | | 1 | | |
| Ambulatory | Yes | 2 | 2.08(1.658-2.49) | 0.00, 0.001 | 0.9412 |
| | No | | 1 | | |
| WHO stages III | Stage III | 3 | 2.98(2.51-3.46) | 0.001, 0.001 | 0.9376 |
| | Stage I & II | | 1 | | |
| WHO stages IV | Stage IV | 7 | 4.17(3.28-5.06) | 84.01, 0.001 | 0.0183 |
| | Stage I & II | | 1 | | |
| Extra PTB | Yes | 8 | 2.56(1.36-3.76) | 95.79, 0.001 | 0.9986 |
| | No | | 1 | | |

study that affect the result of pooled estimate (S10 Fig) with no small study effect (Eggers' p-value=0.9986) (Table 2).

## Discussion

Despite high scale up of anti-TB and antiretroviral therapy, HIV and tuberculosis remain the leading causes of death in TB/ HIV co-infected persons, particularly in middle and low income countries [18, 38]. This systematic review and meta-analysis aimed to estimate the national level pooled mortality rate and its predicting variables using 15 primary studies among TB/ HIV co-infected patients in Ethiopia.

The review finding disclosed that the pooled mortality rate was 18.73% (95% CI: 15.92-20.83). This result was higher than primary studies conducted in Uganda [53], Conakry [54], Nigeria [55], Myanmar [56], South Africa [57] and Botswana [37]. The possible reason might be socio-economic status, study populations, methodological difference (meta-analysis versus primary studies). Most importantly, study time difference because of studies conducted before OVID-19 has low mortality rate than studies done after COVID-19. It is known that the epidemic has interrupted predetermined plans and interventions even the existing ART and anti-TB treatments were endangered during the catastrophic pandemic [58]. However, the finding was lower than the following studies: a systematic review and meta-analyses conducted in low and middle income countries [38], a cohort study conducted in China [18], a finding in Malaysia [20], a randomized trial in Cambodia [59], in Brazil [60] and worldwide systematic review and meta-analysis [7]. The discrepancy might be due to socio-demographic, studies period and culture differences among populations. The other justification might be introduction and strictly adherence of ART and TB treatment guidelines [21].

This systematic review and meta-analysis also revealed that the pooled incidence mortality rate was 4.94 (95% CI: 2.98-6.89). This result is lower than primary studies conducted in

Tanzania [61], Cambodia [59], Uganda [62] and Cameroon [63].While the finding was higher than studies in Nigeria [64] and in South Africa [65]. The plausible reason for this discrepancy might be socio-cultural and economical differences, study time and sample size variation. Additionally, the current study is mixed with adult and children populations and as a result used different methodology might be the reason for inconsistency.

The result of sub-group analysis based on region evidenced that the highest proportion of mortality rate was found in Amhara region followed by Tigray regional state. This might be because of regional variation of care provided for TB/HIV co-infected patients. The other justification would be due to sample size, study follow up time and types of populations studied. The proportion of mortality was higher among patients greater than age 15 and above compared to their counterparts which is supported by the result of systematic review and meta-analysis evidence [15]. This is the fact that opportunistic infection including TB are occur frequently and in severe form in adults which leads to higher proportion of morbidity and mortality [66]. In a similarly way, mortality of patients was high among studies conducted after 2020 as compared to studies done before 2020.This is the fact that after the era of COVID-19, it impose a triple burden on the quality of health service management [67] and has fallen tremendous impact on the control measures of HIV and tuberculosis infections [68, 69].

The current review indicated that patients with poor ART adherence were about four times more likely to die than patients with good adherence status. Studies in Cameroon and Ethiopia [70, 71] had support this evidence. It is kwon that antiretroviral therapy (ART) transforms and improves the life span of persons with HIV and other opportunistic infections by sustain viral suppression and restrain viral replication and, saving the from depletion of the immune system that accelerate the death of patients [72]. No adherence to ART follows drug resistance, treatment failure and exposed the patients to second line ART regimens which results poor patient progress, increase health care costs and reduce survival [73].

TB/HIV co-infected patients with low CD4 count ($<$200ell/mm$^3$) were nearly 3.5 times more hazard to death than patients with normal threshold of CD4 count. This is in agreement with the finding in Tanzania [61], Nigeria [64], Caribbean, Central and South America, and Central, East, and West Africa [74], China [75] and Malaysia [20]. Evidences suggested that HIV causes immunosuppression directly by depletion of host CD4$^+$ T-cells that increase the vulnerability of active TB infection [76] that leads to dramatically fasten the death of persons co-infected with the two worst pandemics [77].

Likewise, patients with low hemoglobin level ($<$10mg/dl) were more likely to die than their counterparts. It is congruent of previous finding of remember clinical trial study [78] and studies in Ethiopia [79]. Anemia in TB/HIV co-infected patients causes treatment failure, lost to follow up and death [80]. It also increase viral load, decrease CD4 count, progress HIV to advanced stage, chronic inflammation and all cause of mortality [81]. Similarly, the current review depicted that TB/HIV co-infected patients with advanced WHO stage were about 4 times hazard to death compared to their counterparts. This is in line with the study done in Tanzania [11, 61] and Malawi [82]. As HIV progress to advanced stage, the patient would be more affected by variety of opportunistic infections, depressed immunity system which further complicated the scenario and decreases survival of TB/HIV co-infected patients [83, 84].

Patients with extra pulmonary tuberculosis were more likely to die than patients with pulmonary tuberculosis which is similar studies in Cameroon [63] and a systematic review elsewhere [85]. Extra pulmonary tuberculosis is the commonest manifestation and leading cause of in HIV-infected patients, particularly in advanced stage of the disease [86]. Extra pulmonary tuberculosis is poorly recognized and which resulted delayed diagnosis and treatment than pulmonary TB and leads to low survival and increase mortality rate [87].

Consistent with the previous literature, this systematic review and meta-analysis also showed that TB/HIV-infected subjects who missing of cotrimoxazole and isoniazid preventing therapy were contribute more hazards to death [88]. Studies reasoned that Isoniazid Preventing Therapy minimize the risk of reactivation of latent TB infection in HIV infected individuals by decreasing mycobacterium load [89]. Cotrimoxazole Preventing Therapy (CPT) also highly prevents suffering and mortality from TB/HIV-co infections [90]. The plausible justification is that CPT can reduce the developments of pneumocystis jirovecii pneumonia which is the main cause of mortality in advanced disease of HIV [91].

The review evidenced that study subjects with bedridden functional status were more hazard to death compared to their counterparts. Bedridden patients were more victimized for immune suppression, severe and complicated opportunistic infections and further deterioration of their health status and fasten the rate of dying [92].

## Strength and limitation of the review

Sensitivity and sub-group analysis was performed to investigate heterogeneity among studies. The included articles with similar study design (cohort) which better identifies cause and effect relationship and reduce methodological variability. Even if the aforementioned strength, the review has its own drawbacks: Qualitative studies and articles published other than English language was excluded. We could not access primary studies from all regions of Ethiopia and some of the included articles also had small sample size that might affect the true pooled estimated result. Due to the nature of meta-analysis, confounding variables that might affect the mortality of TB/HIV co-infected patients were not investigated.

## Conclusion

The analyzed data of this systematic review and meta-analysis depicted that the national pooled proportion and incidence of mortality among TB-HIV co-infected patients were considered to be still high. Being bedridden and ambulatory functional status, poor ART adherence, CD4 count below the threshold ($<200$ cells/mm$^3$), advanced WHO clinical staging, missing of cotrimoxazole and isoniazid preventing therapy, anemia and extra pulmonary TB were significant predictors of mortality. The authors strongly recommended scale up and continuous provision of cotrimoxazole and isoniazid preventive therapy. In addition, early identification and treatment of anemia will greatly halt the high burden of mortality. In general to reduce mortality, morbidity and improve survival, a collaborative effort is mandatory to emphasize close follow up of patients with identified predictors.

## Supporting information

**S1 Checklist. PRISMA 2020 checklist for included studies.**
(DOCX)

**S2 Checklist. JBI critical appraisal checklist for included articles.**
(DOCX)

**S1 Data. Extracted data set from included studies.**
(RAR)

**S1 Table. Search strategies and entry terms from electronic data bases on mortality and its Predictors among TB-HIV co-infected patients in Ethiopia: Systematic and meta-analysis.**
(DOCX)

**S2 Table. List of excluded studies.**
(DOCX)

**S1 Fig. Forest plot of regional sub-group analysis on mortality rate among TB/HIV co-infected patients.**
(TIF)

**S2 Fig. Forest plot of sub-group analysis based on age on mortality rate among TB/HIV co-infected patients.**
(TIF)

**S3 Fig. Forest plot of sub-group analysis based on age on incidence of mortality rate among TB/HIV co-infected patients.**
(TIF)

**S4 Fig. Forest plot of regional sub-group analysis on incidence of mortality rate among TB/HIV co-infected patients.**
(TIF)

**S5 Fig. Forest plot of sub-group analysis according to publication year on incidence of mortality rate among TB/HIV co-infected patients in Ethiopia.**
(TIF)

**S6 Fig. Funnel plot to show publication bias of the included studies.**
(TIF)

**S7 Fig. Leave-one –out sensitivity analysis of CPT on mortality of among TB/HIV co-infected patients.**
(TIF)

**S8 Fig. Leave-one –out sensitivity analysis of factor bedridden on mortality of among TB/HIV co-infected patients.**
(TIF)

**S9 Fig. Leave-one –out sensitivity analysis of WHO stage IV on mortality of among TB/HIV co-infected patients.**
(TIF)

**S10 Fig. Leave-one –out sensitivity analysis of extra PTB on mortality of among TB/HIV co-infected patients.**
(TIF)

## Acknowledgments

The authors are pleased to acknowledge Deber Tabor University for supplementary materials and support to conduct this Review. We also extend our thanks to the authors of included articles in this systematic review and meta-analysis. Last but not least, we acknowledged Bahir Dar University College of health sciences for allowing internet service.

## Author Contributions

**Conceptualization:** Amare Kassaw, Shegaw Zeleke.

**Data curation:** Amare Kassaw, Demewoz Kefale, Tigabu Munye Aytenew, Muluken Chanie Agimas, Shegaw Zeleke, Worku Necho Asferie.

**Formal analysis:** Amare Kassaw.

**Funding acquisition:** Amare Kassaw, Molla Azmeraw.

**Investigation:** Nigatu Dessalegn, Worku Necho Asferie.

**Methodology:** Amare Kassaw, Tigabu Munye Aytenew, Muluken Chanie Agimas.

**Project administration:** Amare Kassaw, Molla Azmeraw, Mastewal Ayehu Sinshaw, Worku Necho Asferie.

**Resources:** Demewoz Kefale, Tigabu Munye Aytenew, Muluken Chanie Agimas, Shegaw Zeleke, Mastewal Ayehu Sinshaw.

**Software:** Amare Kassaw, Demewoz Kefale, Nigatu Dessalegn.

**Supervision:** Molla Azmeraw, Shegaw Zeleke, Mastewal Ayehu Sinshaw.

**Validation:** Molla Azmeraw, Muluken Chanie Agimas, Worku Necho Asferie.

**Writing – original draft:** Amare Kassaw, Demewoz Kefale, Tigabu Munye Aytenew, Molla Azmeraw, Muluken Chanie Agimas, Shegaw Zeleke, Mastewal Ayehu Sinshaw, Nigatu Dessalegn, Worku Necho Asferie.

**Writing – review & editing:** Amare Kassaw, Demewoz Kefale, Tigabu Munye Aytenew, Molla Azmeraw, Muluken Chanie Agimas, Shegaw Zeleke, Mastewal Ayehu Sinshaw, Nigatu Dessalegn, Worku Necho Asferie.

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
