## [Decision Letter · Decision Letter 0]

16 Sep 2024

PONE-D-24-10192Burden of mortality and its Predictors among TB-HIV co-infected patients in Ethiopia: systematic review and Meta-analysis.PLOS ONE

Dear Dr. Kassaw,

Thank you for submitting your manuscript to PLOS ONE. After careful consideration, we feel that it has merit but does not fully meet PLOS ONE’s publication criteria as it currently stands. Therefore, we invite you to submit a revised version of the manuscript that addresses the points raised during the review process.

We look forward to receiving your revised manuscript.

Kind regards,

Musa Mohammed Ali, PhD

Academic Editor

PLOS ONE

Journal Requirements:

2. We note that there is identifying data in the Supporting Information file <S1Table.docx>. Due to the inclusion of these potentially identifying data, we have removed this file from your file inventory. Prior to sharing human research participant data, authors should consult with an ethics committee to ensure data are shared in accordance with participant consent and all applicable local laws.

-Location data

Please remove or anonymize all personal information (Search date), ensure that the data shared are in accordance with participant consent, and re-upload a fully anonymized data set. Please note that spreadsheet columns with personal information must be removed and not hidden as all hidden columns will appear in the published file.

3. As required by our policy on Data Availability, please ensure your manuscript or supplementary information includes the following: 

Reviewers' comments:

Reviewer's Responses to Questions

**Comments to the Author**

1. Is the manuscript technically sound, and do the data support the conclusions?

Reviewer #1: Yes

Reviewer #2: Yes

2. Has the statistical analysis been performed appropriately and rigorously? 

Reviewer #1: Yes

Reviewer #2: Yes

3. Have the authors made all data underlying the findings in their manuscript fully available?

Reviewer #1: Yes

Reviewer #2: Yes

4. Is the manuscript presented in an intelligible fashion and written in standard English?

Reviewer #1: Yes

Reviewer #2: No

5. Review Comments to the Author

Reviewer #1: Some minor error in the construction of paragraphs. The author must revise some repetitions for example 2024. As a systematic review and met analysis I think that all the parts of this types of study has been completed. The aims of the paper has been reached. The introduction gives a summary of the causes of death in the region and globally, specifically for human immunodeficiency infection and tuberculosis. The review disclosed that the pooled proportion and incidence of mortality were 18.73% (95% CI: 15.92-20.83) and 4.94 (95% CI: 2.98-6.89) respectively. Being bedridden and ambulatory functional status, poor ART adherence, CD4 count below the threshold (<200 cells/mm3), advanced WHO clinical staging, not provision of cotrimoxazole and isoniazid preventing therapy, anemia and extra pulmonary TB were significant predictors of mortality. The results reports or are in agreement with the conclusions of the study. It is the first systematic review of its kind done in the country. Should take this in consideration in doing a final assessment of its suitability for acceptance.

Reviewer #2: This study addresses important questions related to the double burden of HIV and TB in Ethiopia. It explores the burden of mortality from HIV and TB co-infection over the past 20 years, offering a better understanding of the predictors of mortality among individuals co-infected with HIV and TB in Ethiopia.

Minor comments

1. Acronyms - The authors can improve clarity by introducing the full term followed by its abbreviation when first mentioned, and then consistently using the acronym in subsequent references, rather than repeating the full term.

2. To ensure consistent formatting, the authors should address some issues. In the discussion section, the word 'REMEMBER' is in capital letters and should be corrected to lowercase. Additionally, 'S7 Figure' is highlighted in red in the results section, which needs adjustment. Lastly, one of the lines in Figure 1 of the results section requires fixing for clarity.

Major comments

1. There appears to be a lack of consistency in the naming of the outcome measures. For instance, the methods section refers to the outcome measures as 'pooled mortality rate' and 'pooled incidence mortality rate,' while the results section uses the term 'incidence density mortality rate.' Additionally, the term 'pooled proportion' is used in the conclusion. It is important to ensure consistency in the terminology used for the outcome measures throughout the manuscript.

It would be helpful to include definitions for the two outcome measures in the methods section and to explain how they were calculated. This could involve specifying both the numerator and denominator for each outcome to ensure clarity and consistency.

2. The manuscript would benefit from thorough proofreading and editing to enhance clarity and accuracy.

3. In the methods section of the abstract, authors may not need to mention the quality assessment of primary studies or publication bias; these details could be reserved for the methods section of the manuscript. Instead, the abstract should include the number of papers included in the systematic review and meta-analysis, the duration of the search period (from 2004 to 2024), and the total sample size. Additionally, specify the analyses performed with the data extracted from the primary studies, including any subgroup analyses conducted by region.

6. PLOS authors have the option to publish the peer review history of their article (what does this mean?). If published, this will include your full peer review and any attached files.

Reviewer #1: **Yes: **Angel Vaillant

Reviewer #2: No

---

## [Author Response · Author response to Decision Letter 0]

26 Sep 2024

Response to Reviewers and editor 

Point by point responses

We would like to take this opportunity to thank the reviewers and editor for sharing their view and constructive comments. The comments were very important which further improves the quality of our manuscript. The point-by-point responses for each of the comments are provided in the following pages. Our responses are written in blue font color.

# Journal requirements

1. Please ensure that your manuscript meets PLOS ONE's style requirements, including those for file naming

Authors’ Response

We are grateful to this comment of technical relevance. Thus, we have ensured that our manuscript meets PLOS ONE's style requirements, including those for file naming

2. We note that there is identifying data in the Supporting Information file <S1Table.docx>. Due to the inclusion of these potentially identifying data, we have removed this file from your file inventory

Authors’ Response 

We accept the comment and remove all personal information (Search date and location) and re-upload the data.

3. As required by our policy on Data Availability, please ensure your manuscript or supplementary information includes the following: 

A numbered table of all studies identified in the literature search, including those that were excluded from the analyses. For every excluded study, the table should list the reason(s) for exclusion. 

Confirmation that the study was eligible to be included in the review 

 This information can be included in the main text, supplementary information, or relevant data repository. Please note that providing these underlying data is a requirement for publication in this journal

Authors’ Response 

Thank you for this concern. We clearly put this concern in figure one that showed all studies identified in the literature search including those that were excluded from the analyses with reasons of exclusion.

Accepting the comment, we have put the extracted data on (S1 data) of supporting information on the revised manuscript. The date and the names of the data extractors were include on the main document of the revised version of this manuscript. The eligibility of the include studies were assessed on the quality assessment part of methods section. 

As detailed in the methods sections, the pooled mortality rate among TB- HIV co-infected patients and its predictors was estimated using random effects model using DerSimonian-Laird model weight by utilizing Stata 17 statistical software. 

Regarding to how missing data were handled, an explanation is done on the revised manuscript (page 6). 

4. Please review your reference list to ensure that it is complete and correct. If you have cited papers that have been retracted, please include the rationale for doing so in the manuscript text, or remove these references and replace them with relevant current references. Any changes to the reference list should be mentioned in the rebuttal letter that accompanies your revised manuscript. If you need to cite a retracted article, indicate the article’s retracted status in the References list and also include a citation and full reference for the retraction notice

Authors’ Response

The authors are very grateful of these constructive comments. After read the reference lists carefully, we made some correction and addressed the incomplete one. We are sure that we have not cited retracted references, almost all references are recent. 

Response to Reviewer #1 comments 

5. Some minor error in the construction of paragraphs. The author must revise some repetitions for example 2024. As a systematic review and meta-analysis I think that all the parts of this types of study has been completed…..the aims of the paper have been reached the results reports or are in agreement with the conclusions of the study. It is the first systematic review of its kind done in the country. Should take this in consideration in doing a final assessment of its suitability for acceptance.

Authors’ Response 

First of all, we really thank the reviewer for his constructive comments and suggestions. The authors have accepted the comments and corrected it in the revised version of the manuscript

Reviewer #2: 

Minor comments

1. Acronyms - The authors can improve clarity by introducing the full term followed by its abbreviation when first mentioned, and then consistently using the acronym in subsequent references, rather than repeating the full term.

Authors’ Response 

Dear reviewer, we want to say thank you very much for your relevant, constructive and detail comments and suggestions that have a great value to improve our manuscript. The comment is taken and corrected accordingly. 

2. To ensure consistent formatting, the authors should address some issues. In the discussion section, the words ‘REMEMBER' is in capital letters and should be corrected to lowercase. Additionally, 'S7 Figure' is highlighted in red in the results section, which needs adjustment. Lastly, one of the lines in Figure 1 of the results section requires fixing for clarity

Authors’ Response 

Thank you and we corrected it.

Major comments

1. There appears to be a lack of consistency in the naming of the outcome measures. For instance, the methods section refers to the outcome measures as 'pooled mortality rate' and 'pooled incidence mortality rate,' while the results section uses the term 'incidence density mortality rate.' Additionally, the term 'pooled proportion' is used in the conclusion. It is important to ensure consistency in the terminology used for the outcome measures throughout the manuscript.

It would be helpful to include definitions for the two outcome measures in the methods section and to explain how they were calculated. This could involve specifying both the numerator and denominator for each outcome to ensure clarity and consistency.

Authors’ Response 

Based on the comment, correction is made on the revised version of the manuscript. 

2. The manuscript would benefit from thorough proofreading and editing to enhance clarity and accuracy.

Authors’ Response

Thank you for your constructive comments. After we have read carefully through the whole document, we properly addressed the concerned issues. 

Accepting the comment, the authors have read thoroughly and edited carefully the whole manuscript before submission.

3. In the methods section of the abstract, authors may not need to mention the quality assessment of primary studies or publication bias; these details could be reserved for the methods section of the manuscript. Instead, the abstract should include the number of papers included in the systematic review and meta-analysis, the duration of the search period (from 2004 to 2024), and the total sample size. 

Authors’ Response

Accepting the comment, the authors have corrected accordingly. 

4. Additionally, specify the analyses performed with the data extracted from the primary studies, including any subgroup analyses conducted by region.

Authors’ Response

After we extracted the data from primary studies on Microsoft excel, the analysis was performed by using STATA 17 statistical software. Then, the pooled mortality rate among TB- HIV co-infected patients and its predictors was estimated using random effects model using DerSimonian-Laird model weight.

Sub-group analysis was also performed using publication year, region, sample size and age of patients from the extracted data and presented using frost plot.

---

## [Decision Letter · Decision Letter 1]

11 Oct 2024

Burden of mortality and its Predictors among TB-HIV co-infected patients in Ethiopia: systematic review and Meta-analysis.

PONE-D-24-10192R1

Dear Dr. Kassaw,

We’re pleased to inform you that your manuscript has been judged scientifically suitable for publication and will be formally accepted for publication once it meets all outstanding technical requirements.

Kind regards,

Musa Mohammed Ali, PhD

Academic Editor

PLOS ONE

Additional Editor Comments (optional):

Reviewers' comments:

Reviewer's Responses to Questions

**Comments to the Author**

1. If the authors have adequately addressed your comments raised in a previous round of review and you feel that this manuscript is now acceptable for publication, you may indicate that here to bypass the “Comments to the Author” section, enter your conflict of interest statement in the “Confidential to Editor” section, and submit your "Accept" recommendation.

Reviewer #1: All comments have been addressed

Reviewer #2: All comments have been addressed

2. Is the manuscript technically sound, and do the data support the conclusions?

Reviewer #1: Yes

Reviewer #2: Yes

3. Has the statistical analysis been performed appropriately and rigorously? 

Reviewer #1: Yes

Reviewer #2: Yes

4. Have the authors made all data underlying the findings in their manuscript fully available?

Reviewer #1: Yes

Reviewer #2: Yes

5. Is the manuscript presented in an intelligible fashion and written in standard English?

Reviewer #1: Yes

Reviewer #2: (No Response)

6. Review Comments to the Author

Reviewer #1: The authors have carefully addressed all the comments raised during the peer review process. After thorough revisions, the research work appears accurate and unbiased.

Reviewer #2: (No Response)

7. PLOS authors have the option to publish the peer review history of their article (what does this mean?). If published, this will include your full peer review and any attached files.

Reviewer #1: **Yes: **Dr. Angel Vaillant

Reviewer #2: No

---

## [Editor Report · Acceptance letter]

24 Oct 2024

PONE-D-24-10192R1 

PLOS ONE

Dear Dr. Kassaw, 

I'm pleased to inform you that your manuscript has been deemed suitable for publication in PLOS ONE. Congratulations! Your manuscript is now being handed over to our production team.

Kind regards, 

on behalf of

Dr. Musa Mohammed Ali 

Academic Editor

PLOS ONE